# AdaRec: Adaptive Recommendation for Reinforcing Long-term User Engagement

## Abstract

Growing attention has been paid to Reinforcement Learning (RL) algorithms when optimizing long-term user engagement in sequential recommendation tasks. One challenge in large-scale online recommendation systems is the constant and complicated changes in users' behavior patterns, such as interaction rates and retention tendencies. When formulated as a Markov Decision Process (MDP), the dynamics and reward functions of the recommendation system are continuously affected by these changes. Existing RL algorithms for recommendation systems will suffer from distribution shift and struggle to adapt in such an MDP. In this paper, we introduce a novel paradigm called Adaptive Sequential Recommendation (AdaRec) to address this issue. AdaRec proposes a new distance-based representation loss to extract latent information from users' interaction trajectories. Such information reflects how RL policy fits to current user behavior patterns, and helps the policy to identify subtle changes in the recommendation system. To make rapid adaptation to these changes, AdaRec encourages exploration with the idea of optimism under uncertainty. The exploration is further guarded by zero-order action optimization to ensure stable recommendation quality in complicated environments. We conduct extensive empirical analyses in both simulator-based and live sequential recommendation tasks, where AdaRec exhibits superior long-term performance compared to all baseline algorithms.

## 1 Introduction

Recent sequential recommendation algorithms have achieved preliminary success in optimizing long-term user engagement with the assistance of Reinforcement Learning (RL) (Zou et al., 2019; Xue et al., 2023a; Cai et al., 2023a). Long-term engagement is considered more desirable than immediate feedback as it is directly linked to practical metrics such as daily active users (DAU). RL (Sutton & Barto, 1998) is well suited to optimize long-term engagement because it can efficiently handle delayed reward signals (Sutton, 1992) and facilitate efficient exploration (Ciosek et al., 2019). However, large-scale online recommendation platforms in the real world can exhibit constant changes in users' behavior patterns and their willingness of long-term engagement. For example, stock traders may be more willing to get back to a news recommendation application when there are exciting financial news they care about. This phenomenon will lead to evolving dynamics and reward functions of the Markov Decision Process (MDP) in the sequential recommendation task. Therefore, the complexity of real-world recommendation systems calls for algorithms that can identify environment changes and make rapid adaptation.

Unfortunately, the issue of evolving user behavior patterns has largely been overlooked by current state-of-the-art RL algorithms for recommendation systems (Zhang et al., 2022; Xue et al., 2023a; Cai et al., 2023b;a). When these algorithms are deployed in real-world tasks, recommendation agents will suffer from distribution shift between different trajectories, leading to unstable training and poor practical performance. Although several RL algorithms have been proposed to address the issue of distribution shift, optimizing long-term user engagement in sequential recommendation poses unique challenges that current methods struggle to address. For example, algorithms focused on representation learning (Zhang et al., 2021; Mazoure et al., 2022) aim to learn state representations that are insensitive to visual disturbances in policy inputs. But in sequential recommendation the policy inputs are dense features of users and videos, and the disturbances occur in the environment dynamics. Other algorithms in Meta-RL or zero-shot policy generalization (Rakelly et al., 2019; Luo et al., 2022) attempt to explicitly identify environment parameters (e.g., robot arm masses or joint

frictions), as additional policy input. But several factors that influence user behavior in sequential recommendation are interrelated, making it challenging to explicitly represent each behavior pattern with the environment parameters.

In this paper, we first perform data-driven analyses on the open-source KuaiRand dataset (Gao et al., 2022) for unbiased sequential recommendation. Through the analyses, we observe that in different time periods, user behaviors indeed exhibit different patterns in terms of preferences, frequencies of immediate feedback, and the distribution of return time. We then propose a novel paradigm called Adaptive Sequential Recommendation (AdaRec) for training adaptive policies. AdaRec introduces a context encoder in the policy network that enables RL policies to identify and adapt to different user behavior patterns. The encoder is trained by a distance-based loss, minimizing the discrepancy between the $l_2$-distance in the encoder output space and a performance-related distance measure in the state space. To make rapid adaptation to environment changes, AdaRec also encourages exploration with the idea of optimism under uncertainty. The exploration is further guarded by zero-order action optimization to ensure a stable recommendation quality in complicated environments.

To evaluate AdaRec in optimizing long-term user engagement, we conduct experiments in both the KuaiSim user retention simulator (Zhao et al., 2023) and a real-world short video recommendation platform, which supports continuous experiments for several weeks involving millions of users and billions of interactions. By manually altering the simulator feedback in each episode, we manage to simulate the evolving user behaviors that occur in practical scenarios. Experimental results in the modified simulator and the online platform demonstrate that AdaRec surpasses state-of-the-art methods in terms of both training stability and adaptation ability in complex environments with distribution shift.

## 2 BACKGROUND

### 2.1 PRELIMINARIES

The sequential recommendation problem can be represented as a Markov Decision Process (MDP), defined by the tuple $< \mathcal{S}, \mathcal{A}, T, r, \gamma >$. In this formulation, $\mathcal{S}$ represents the state space, $\mathcal{A}$ denotes the action space, $T$ is the transition function, $r$ corresponds to the reward function, and $\gamma$ is the discount factor. Figure 1(a) illustrates the connection between the MDP formulation and the actual recommendation procedure. In this framework, users are treated as environments and the recommendation model operates by taking actions within this environment. At timestep $t$, the state $s_t$ is formed by incorporating various features, including user profile, user interaction history, and candidate item profiles. A deep scoring model is used to predict a $k$ dimensional score $x_i = (x_{i1}, x_{i2}, \cdots, x_{ik})$ for each selected item $i$, where each dimension evaluates the item in a particular aspect. Details of the scoring model can be found in the literature (Cai et al., 2023a) and will be treated as a black box in this paper. The deterministic RL policy $\pi$ is responsible for generating a $k$-dimensional action $a_t = \pi(s_t)$. Subsequently, a pre-defined ranking function $f$ is employed to compute the final ranking score $f(a_t, x_i)$ for each selected item $i$. The system then recommends the top-$n$ items to the user. In this paper, we set $k = 7$ and the action space is a 7-dimensional continuous space. The immediate feedback provided by the user is utilized to calculate the immediate reward $r_t^{im}$. The interaction information is used to update the user profile and browsing history, resulting in a deterministic transition function $T$. The episode concludes when the user leaves. When the same user returns, the time of their return is used to calculate the delayed retention reward $r_t^{re}$. The reward function $r(s_t, a_t)$ is a linear combination of the immediate and delayed rewards.

RL aims at maximizing the accumulated return of the policy $\pi$: $\eta_T(\pi) = E_{\pi,T}[\sum_{t=0}^{\infty} \gamma^t r(s_t, a_t)]$, where the expectation is computed with $a_t \sim \pi(\cdot|s_t)$, and $s_{t+1} \sim T(\cdot|s_t, a_t)$. In an MDP with a policy $\pi$, the state-action value function $Q_T^\pi(s, a)$ denotes the expected return after taking action $a$ at state $s$: $Q_T^\pi(s, a) = E_{\pi,T} \sum_{t=0}^{\infty} \gamma^t r(s_t, a_t)|s_0 = s, a_0 = a]$. It is also referred to as the Q-function. The state value function, or the V-function, is defined as $V_T^\pi(s) = \mathbb{E}_{a \sim \pi(\cdot|s)} Q_T^\pi(s, a)$.

### 2.2 RELATED WORK

**RL for Sequential Recommendation** In Reinforcement Learning (RL) (Sutton & Barto, 1998), a learning agent interacts with the environment (Silver et al., 2017) or exploits offline dataset (Fujimoto et al., 2019) to optimize the cumulative reward obtained throughout a trajectory. RL is particularly well-suited for sequential recommendation tasks, where users are considered as environments and

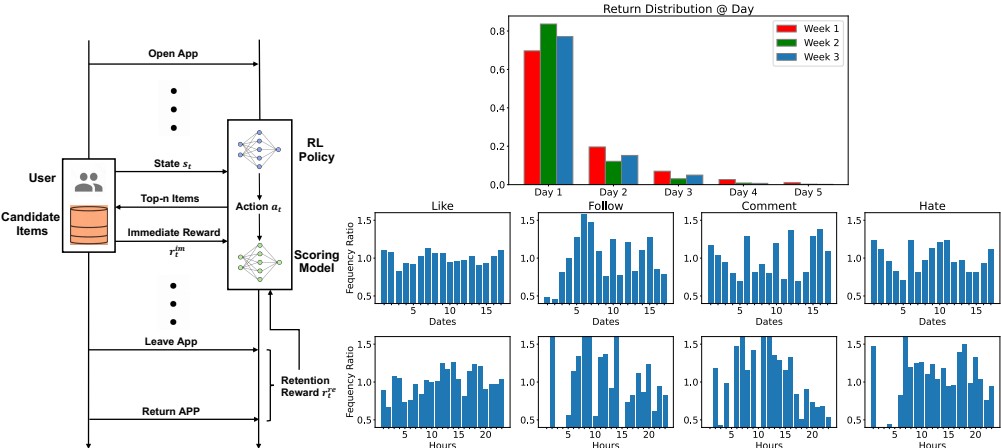

Figure 1: **Left:** The MDP for optimizing long-term user engagement with Reinforcement Learning. **Upper Right:** The distribution of user return time in three weeks. **Lower Right:** Normalized frequencies of immediate user feedback, compared on different dates in one month and different hours in one day.

immediate user feedback is utilized to compute the reward (Shani et al., 2005; Chen et al., 2018; Zhang et al., 2022). Traditional approaches propose to learn an additional world model (Bai et al., 2019) or explicitly construct high-fidelity simulators (Shi et al., 2019) to enhance the sample efficiency of RL algorithms. Recently, increasing attention has been paid to optimizing long-term user retention with RL (Zou et al., 2019). RLUR (Cai et al., 2023a) comprehensively deals with issues that comes along in this field, including delayed reward, uncertainty, and instability of training. Other approaches focus on reward engineering, employing the constrained actor-critic method (Cai et al., 2023b) or incorporating human preferences (Xue et al., 2022). Our paper also aims at optimizing long-term user engagement and focuses on mitigating the challenge of distribution shift that has largely been ignored in previous methods.

**Adaptive Policy Training with RL**   There are a handful of RL algorithms that train adaptive policies in evolving environments with distribution shift. Meta-RL algorithms (Rakelly et al., 2019; Beck et al., 2023) can adapt to new environments by fine-tuning a small amount of data from the test environment. Zero-shot adaptation algorithms like ESCP (Luo et al., 2022) and SRPO (Xue et al., 2023b) can fit new environments without additional data. A key component enabling rapid adaptation of RL policies is the context encoder (Xu et al., 2021), which takes a stack of history states as input and generates a dense vector that represents different contexts. The encoder can be trained with variational inference (Zintgraf et al., 2020) or auxiliary losses (Luo et al., 2022). Its output can be used as additional inputs to the environment model (Lee et al., 2020) or the policy network (Chen et al., 2021). Other approaches focus on representation learning (Zhang et al., 2021) or importance sampling (Liu et al., 2022). However, none of these methods specifically consider the practical requirements of sequential recommendation tasks and may be unreliable or inefficient when applied directly. In our proposed approach, we introduce a new distance-based loss to train the context encoder that is directly related to the estimated length of user retention.

We leave relevant researches on recommendation with evolving user interests in Appendix B.

## 3   EVOLVING USER BEHAVIORS IN SEQUENTIAL RECOMMENDATION TASKS

The main focus of this paper is to address the issue of distribution shift arisen from evolving patterns of user behaviors in sequential recommendation. To empirically justify the existence of such issue in practical recommendation systems, we conduct a data-driven study on the KuaiRand dataset (Gao et al., 2022)[1], which is a comprehensive and unbiased sequential recommendation dataset. It collects the recommendation logs of a popular video-sharing mobile application from 8 April to 8 May 2022,

---

[1]In this paper, we choose the KuaiRand-Pure branch.

involving 27,285 users, 7,551 items, and 188,562 interactions. To avoid the influence of different interaction histories, we select interactions that happen at the start of each recommendation session, i.e., the $s_0$ of the trajectory. Regarding the three aspects of user behaviors, the user preferences are extensively studied in previous methods (Zhao et al., 2020; 2021) and not related to our formulation. For the purposes of this paper, we are primarily interested in investigating the users' return-time distribution and frequencies of immediate interactions, including like, follow, comment, and hate.

In Fig. 1 (right), we visualize the distribution of the user return time in three weeks, as well as the normalized interaction rates between different dates and hours. We observe that the return probability and interaction frequency exhibit variability over time in a range from 10% to 50%, depending on the type of feedback. For example, the average probability of user returning to the application in the next day can be as low as about 70% in week 1, and as high as about 81% in week 2. Among immediate signals, the "like" signal exhibits relatively more stability, while the other three signals deviate significantly from the average. For example, the "follow" signal is about twice more frequent on day 5 than on day 9. Furthermore, we can hardly identify any clear pattern in the changes of interaction frequency between dates within a month or hours within a day. Hence, it is hard to manually extract environmental information with predefined rules and feed it to the policy network.

The aforementioned fluctuations in user return probability and interaction frequency will significantly influence the Markov Decision Process (MDP) that describes the task of sequential recommendation. To optimize long-term user engagement, the reward function $r(s_t, a_t)$ of the MDP is related to the users' return-time distribution. As previously discussed, given the same distribution of states $s_t$ and actions $a_t$, the users' return time exhibits fluctuations across different weeks. This implies that the reward distribution $R(r|s_t, a_t)$ on $(s_t, a_t)$ is time-variant, i.e., it depends on the timestep $t$. Meanwhile, the history of immediate user feedback is incorporated in the state space $\mathcal{S}$ of the MDP. When the same pair of state-action $(s_t, a_t)$ is considered, variations in feedback at timestep $t$ result in a different distribution of subsequent state $s_{t+1}$. This leads to time-varying environment dynamics $T(s_{t+1}|s_t, a_t)$, which is also influenced by evolving user behavior patterns. Such time-variant environment dynamics and reward functions give rise to the issue of distribution shift in training data collected at different timesteps, which current recommendation algorithms in RL struggle to handle (Chen et al., 2021; Xue et al., 2023b).

## 4 ADAREC: ADAPTIVE SEQUENTIAL RECOMMENDATION

Analyses in the previous section have highlighted the continuous changes in the MDPs describing sequential recommendation tasks. In such MDPs, the RL model faces two major challenges to produce high-quality recommendations: accurately identifying different user behavior patterns and rapidly adapting to sudden pattern changes. To address the first challenge, we incorporate context encoders into the policy and value network. These encoders are trained using a specific distance-based loss, and is capable of detecting transitions in user behavior patterns and notifying the learning policy accordingly. To achieve rapid policy adaptation, we encourage exploration during training with

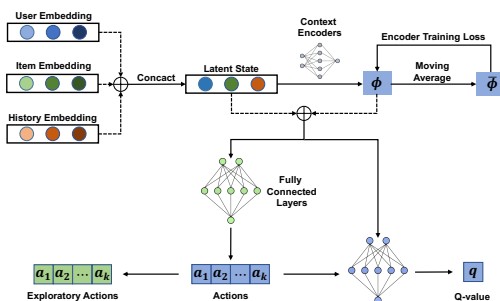

Figure 2: The framework of AdaRec.

the idea of optimism under uncertainty. Additionally, we introduce an extra action optimization step to ensure performance stability. Summarizing these contributions, we propose the AdaRec (**Ada**ptive Sequential **Rec**ommendation) algorithm. Its architecture is shown in Fig. 2 and the algorithm procedure is listed in Appendix C. We discuss in detail the mechanism of encoder training and optimistic exploration in Sec. 4.1 and Sec. 4.2, respectively.

### 4.1 IDENTIFYING USER BEHAVIOR PATTERNS WITH CONTEXT ENCODERS

To identify changes in dynamic environments and address the challenge of distribution shift, previous methods employ context encoders to identify latent environment parameters that directly capture

the evolution of environments (Lee et al., 2020; Luo et al., 2022). For example, in the autonomous driving scenario with an evolving road friction coefficient (Li et al., 2023), the encoder output should align with the changes in friction. However, in the context of sequential recommendation, a variety of latent factors can collaboratively influence how the environment changes. Users will exhibit different behavior patterns on various dates of the month, days of the week, hours of the day, and in response to different social media trends, among other factors. Therefore, it is difficult, if not impossible, to explicitly identify environment parameters that correspond to environment changes.

Instead of expecting context encoders to generate environment parameters, in this paper we propose to directly relate encoder outputs to the expected performance of the current policy. When the performance is degraded, the policy can adapt accordingly and adjust output patterns. To achieve this, we enforce the $l_2$-distance in the encoder output space to be close to a specific distance measure $d$ that reflects policy performance. The loss function for updating the context encoder $\phi$ can then be expressed as

$$J(\phi) = \sum_{i,j} \left[ \left\| \phi\left(s_i\right) - \phi\left(s_j\right) \right\|_2 - d\left(s_i, s_j\right) \right]^2. \tag{1}$$

In the Actor-Critic architecture of RL, the state value function $V$ can serve as the critic and can evaluate the performance of the learning policy. If both two states $s_i, s_j$ have high state values, the policy will perform well on both of them, so the latent variable $\phi(s_i)$ should be close to $\phi(s_j)$. If two states have different state values, their corresponding latent variables should be far from each other. Therefore, we choose the following distance measure based on the state value function $V$:

$$d\left(s_i, s_j\right) = \begin{cases} 0 & \text{if } V\left(s_i\right) \text{ and } V\left(s_j\right) \text{ are close,} \\ \infty & \text{otherwise.} \end{cases} \tag{2}$$

To determine whether $V(s_i)$ and $V(s_j)$ are close, we rank a batch of input states $s_1, s_2, \cdots, s_B$ with size $B$ by their state values and divide them into $n$ categories $C_1, C_2, \cdots, C_n$, where $n$ is a hyperparameter. States that are assigned to the same category are considered to have similar values. We denote $j \in N(i)$ if $s_i$ and $s_j$ fall into the same category. Plugging Eq. (2) into the original loss function Eq. (1), we get[2]

$$J(\phi) = \sum_i \left[ \sum_{j \in N(i)} \left\| \phi\left(s_i\right) - \phi\left(s_j\right) \right\|_2^2 - \sum_{j \notin N(i)} \left\| \phi\left(s_i\right) - \phi\left(s_j\right) \right\|_2^2 \right], \tag{3}$$

where the first term makes encoder outputs of states in the same category closer, and the second term pushes states in different categories away from each other.

In practice, the state-value function $V$ is updated alongside policy training and the state batch is randomly sampled from the replay buffer that keeps updating. Therefore, the output $\phi(s_i)$ can be unstable, which is undesirable when using $\phi(s_i)$ as part of the policy input. To mitigate this problem, we incorporate the moving average $\tilde{\phi}_k = (1 - \eta)\tilde{\phi}_k + \frac{\eta}{|C_k|} \sum_{s_i \in C_k} \phi(s_i), k = 1, 2, \cdots, n$ of each state categories to both terms in Eq. (3). We transform Eq. (3) to make it related with the average encoder output $\bar{\phi}_k = \frac{1}{|C_k|} \sum_{s_i \in C_k} \phi(s_i)$. With regard to the first term (denoted as $J_{\text{same}}(\phi)$), we have

$$J_{\text{same}}(\phi) = \sum_{k=0}^n \sum_{s_i, s_j \in C_k} \left\| \phi\left(s_i\right) - \phi\left(s_j\right) \right\|_2^2 = \frac{2B}{n} \sum_{k=0}^n \sum_{s_i \in C_k} \left\| \phi(s_i) - \bar{\phi}_k \right\|_2^2. \tag{4}$$

To maximize $J(\phi)$ with states from different categories (denoted as $J_{\text{diff}}(\phi)$), we have

$$J_{\text{diff}}(\phi) = -\sum_{k=0}^n \sum_{m>k} \sum_{s_i \in C_k} \sum_{s_j \in C_m} \left\| \phi\left(s_i\right) - \phi\left(s_j\right) \right\|_2^2 \leqslant -\frac{B^2}{n^2} \sum_{k=0}^n \sum_{m>k} \left\| \bar{\phi}_k - \bar{\phi}_m \right\|_2^2. \tag{5}$$

By minimizing the last term in Eq. (5), we are minimizing an upper-bound of the original loss function. The average encoder output $\bar{\phi}_k$ can be replaced with the moving average $\tilde{\phi}_k$, giving rise to the final loss function that is used during training:

$$J(\phi) = \frac{2B}{n} \sum_{k=0}^n \sum_i \left\| \phi(s_i) - \tilde{\phi}_k \right\|_2^2 - \frac{B^2}{n^2} \sum_{k=0}^n \sum_{m>k} \left\| \tilde{\phi}_k - \tilde{\phi}_m \right\|_2^2. \tag{6}$$

---

[2]The detailed derivations in this section are listed in Appendix A.

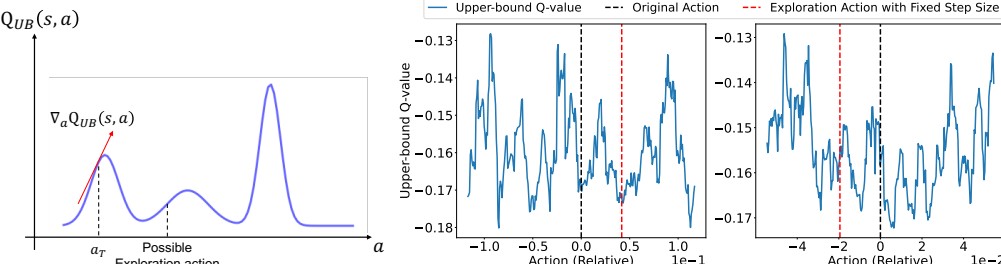

Figure 3: **Left:** The demonstration of the gradient-based exploration action selection with optimistic state-action values. The action can miss the local optimums of the state-action value function; **Right:** The state-action value function on two example state-action pairs. The vertical lines show the relative values of the original and exploration action in one dimension. The exploration action generated with a fixed step size can lead to a lower value than the original action.

In practice, in addition to the distance-based loss function $J(\phi)$, the policy optimization loss is also backpropagated to the context encoder during training to accelerate the training process. As demonstrated in Fig. 2, the encoder output $\phi(s)$ is also used as the input to the state-value function $V$ for a more accurate value estimation.

## 4.2 ADAPTING TO PATTERN CHANGES WITH OPTIMISTIC EXPLORATION

Previously we discussed how context encoders can identify specific patterns of the user behavior using latent variables $\phi(s)$. These latent variables are concatenated with the original state $s$ and fed into the policy network $\pi$, which outputs the action $a_T = \pi(s, \phi(s))$. When representing two distinct behavior patterns, we expect $\phi(s)$ to exhibit significant differences between them. This increases the diversity of inputs to the policy network and necessitates a larger amount of training data. To improve the training efficiency, we explore the concept of optimism in face of uncertainty, which encourages the agent to engage in exploration.

Instead of evaluating the state-action pair $(s, a)$ pessimistically with the minimum of a pair of Q networks (Fujimoto et al., 2018; Haarnoja et al., 2018), the optimistic state-action value estimation is defined as $Q_{\text{UB}}(s, a) = \mu_Q(s, a) + \beta\sigma_Q(s, a)$, where $\mu_Q$ and $\sigma_Q$ represent the mean and standard deviation of the outputs from the Q networks, respectively, and $\beta$ is a hyper-parameter. According to (Ciosek et al., 2019), the exploration action $a_E$ can be calculated by extending the original policy output $a_T$ in the direction of gradient ascent of $Q_{\text{UB}}(s, a)$: $a_E = a_T + \delta \cdot \frac{\Delta}{\|\Delta\|}$, where $\Delta = [\nabla_a Q_{\text{UB}}(s, a)]_{a=a_T}$ is the gradient of $Q_{\text{UB}}$ with respect to the action, and $\delta$ is the step size hyperparameter. However, determining the extent to which the original action should be extended, i.e., the step size $\delta$, can be challenging, as a small step size may lead to inefficient exploration, while a large step size can result in inaccurate linear approximation. As illustrated in Fig.3 (left), the state-action function can exhibit multiple peaks, and an improper step size may cause the exploration action to miss a local optimum, resulting in poorer exploration.

The challenge of selecting an appropriate step size $\delta$ is more pronounced in sequential recommendation tasks, as the landscape of state-action values in such tasks can exhibit high complexity, which is illustrated in Fig. 3 (middle and right). This complexity arises from the presence of numerous peaks and significant oscillations. Consequently, it becomes difficult to identify a fixed step size that consistently performs well throughout the training process. Another reason is that recommendation tasks can be risk-sensitive: users may disengage from the application and cease the recommendation process if they encounter recommended items that fail to capture their interest. To mitigate the aforementioned issues, we use the practice in zero-order optimization. A set of actions $a_k, k = 1, 2, \cdots, n$ are located near the original policy output $\pi(s, \phi(s))$, in the direction of the gradient $\nabla_a Q_{\text{UB}}(s, a)$. The exploration action $a_E$ is sampled form the action set in a softmax manner:

$$p(a_k|s) \propto \exp(Q_{\text{UB}}(s, a_T + k\delta\left[\nabla_a Q_{\text{UB}}(s, a)\right]_{a=\pi(s)})), \tag{7}$$

where $\delta$ is the hyper-parameter controlling the gap of action particles. It can be set to a small value and does not need extra tuning. By choosing from several candidate actions, the exploration module manages to find actions with higher state-action values more efficiently and reduces the risk of

adopting dangerous actions that have low values. We demonstrate the effectiveness of this exploration technique in Sec. 5.4.

## 5 EXPERIMENTS

To evaluate and analyse the practical performance of AdaRec, we conduct extensive experiments in the task of short video recommendation to investigate the following research questions (RQs): **RQ1**: Can the framework of AdaRec lead to performance improvement in long-term user engagement when applied to environments with distribution shift? **RQ2**: How does each component of AdaRec contribute to overall performance? **RQ3**: Can AdaRec perform well in online A/B tests of large-scale live recommendation platforms? To answer these questions, AdaRec is first used to generate recommendation policies in the KuaiSim retention simulator (Zhao et al., 2023) with a manually designed level of distribution shift. It is then deployed in a dynamic, large-scale, real-world short video recommendation platform to perform live A/B test. We also conduct ablation studies and visualizations to investigate the contribution of each component of AdaRec to the overall performance.

### 5.1 SETUP

**Simulator**    AdaRec adopts a novel paradigm that focuses on optimizing long-term user engagement, rather than immediate user feedback. However, recommendation simulators that have been widely used (Shi et al., 2019; Ie et al., 2019; Wang et al., 2021) cannot simulate long-term user behaviors, such as the probability of returning to the application in a few days. Instead, we select the KuaiSim retention simulator (Zhao et al., 2023) that aims to simulate long-term user behavior on short video recommendation platforms. It has been used by various recommendation algorithms (Liu et al., 2023a;b; Cai et al., 2023a) that also investigate long-term user behaviors. The KuaiSim simulator contains a leave module, which predicts whether the user will leave the session and terminate the episode; and a return module, which predicts the probability of the user returning to the platform on each day as a multinomial distribution. The average user return time is used to calculate the reward $r$ of an episode. As AdaRec is designed to handle the issue of distribution shift, we manually alter the probabilities of user leaving and returning by up to 20% in each episode. This allows us to capture the dynamic nature of user behaviors. More information on the simulator setup is in Appendix D.

**Baselines**    In simulator-based experiments, we compare AdaRec with various baselines, such as 1) state-of-the-art value-based RL algorithms, including TD3 (Fujimoto et al., 2018) and SAC (Haarnoja et al., 2018); 2) RL algorithms that facilitate efficient exploration, including OAC (Ciosek et al., 2019) and RND (Burda et al., 2019); 3) a context encoder-based RL algorithm ESCP (Luo et al., 2022); 4) an RL-based recommendation algorithm for optimizing long-term user engagement RLUR (Cai et al., 2023a); 5) non-RL recommendation method, including CEM (Deng, 2006) and DIN (Zhou et al., 2018). We did not compare with few-shot Meta-RL methods (Rakelly et al., 2019) because the recommendation system changes constantly and unpredictably. It is impossible to obtain trajectories in each target environment for policy finetuing before deployment. Previous non-RL recommendation algorithms that take evolving user interests into account (Brown & Agarwal, 2022; Zhao et al., 2020; 2021) are not included, as they all focus on immediate feedback and cannot naturally fit to recommendation tasks that optimize the long-term user experience. In live experiments, we only select TD3, ESCP, and RLUR as baseline algorithms due to the potential negative impact of suboptimal policies. Details of the baseline algorithms are described in Appendix D.

**Evaluation Metrics**    In simulator-based experiments, we choose three criteria to evaluate the algorithms: the users' average return days (the lower the better), the users' return probability on the next day (the higher the better), and the cumulative retention reward (the higher the better). All algorithms for comparison are run for 50,000 training steps with five different random seeds. In live experiments, a crucial metric we use is the rate of users returning to the platform in 7 days, which is in accordance with the goal of maximizing long-term user retention. We also focus on the application dwell time, as well as immediate user responses including video click-through rate (CTR) (click and watch the video), like (like the video), comment (provide comments on the video), and unlike (unlike the video). These metrics are standard evaluation criteria for recommendation algorithms and are empirically shown to be related to long-term user experiences (Wang et al., 2022).

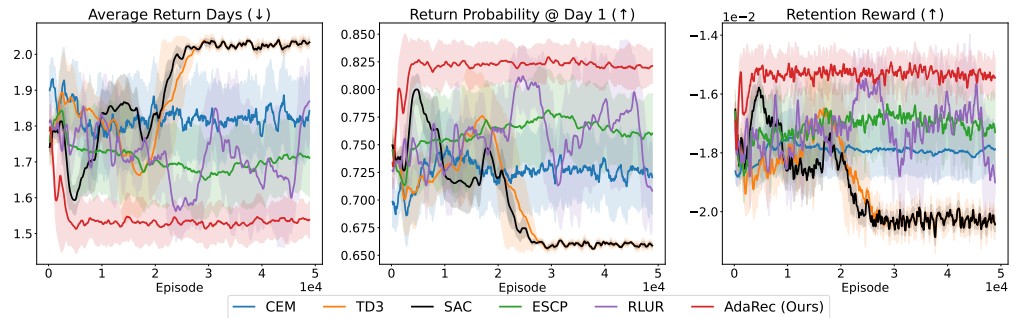

Figure 4: Performance comparison of different algorithms in the modified KuaiSim simulator. Metrics with the up arrow (↑) are better with larger values and vice versa.

Table 1: Performance comparison of different algorithms with the RLUR baseline in live experiments. Metrics with the up arrow (↑) are better with larger values and vice versa.

|  | 7d Retention Rate ‰, ↑ | Dwell Time ‰, ↑ | Click-through Rate ‰, ↑ | Like Rate ‰, ↑ | Comment Rate ‰, ↑ | Unlike Rate %, ↓ |
|---|---|---|---|---|---|---|
| TD3 | 0.041±0.148 | -0.685±0.297 | 1.412±0.619 | 1.798±1.127 | 1.715±1.782 | -0.567±0.820 |
| ESCP | 0.123±0.073 | -0.115±0.277 | 0.703±0.313 | 0.302±0.952 | -1.975±1.204 | -0.116±0.791 |
| AdaRec (Ours) | **0.138**±0.089 | **0.263**±0.181 | **3.260**±0.332 | **2.821**±0.925 | **8.392**±1.881 | **-1.874**±0.781 |

## 5.2 PERFORMANCE COMPARISON IN SIMULATOR

The training curve for AdaRec, as well as baseline algorithms CEM, TD3, SAC, ESCP, and RLUR are shown in Fig. 4. The table for comparisons with all baseline algorithms is in Appendix D. The CEM algorithm performs worse than the other RL-based algorithms. This highlights the effectiveness of RL in optimizing long-term user engagement. The performance of TD3 and SAC exhibits improvements in the early stage of training, but deteriorates as training proceeds. Without explicit modeling of the environment distribution shift, the policies they obtain are loosely coupled with specific user behavior patterns, leading to suboptimal performance. The RLUR algorithm takes into account the bias of the sequential recommendation task and outperforms TD3 and SAC. But it suffers from unstable training and takes more steps to converge than AdaRec. The ESCP algorithm incorporates a context encoder that can capture the distribution shift to some extent. But it explicitly relies on a single environment parameter, and cannot model environment changes thoroughly. As a result, it has a stable training curve, but exhibits suboptimal overall performance. Compared with baseline algorithms, AdaRec shows a stable training curve and the best overall performance. The stability is due to the context encoder module which enables the algorithm to fit different environment dynamics and reward functions. The good asymptotic performance can be attributed to the safe and efficient exploration module that quickly navigates to high-reward regions when the environment changes.

## 5.3 LIVE EXPERIMENTS

The MDP setup in the live experiments is similar to that described in Sec. 2.1. The algorithms are incorporated in a candidate-ranking system of a popular short-video recommendation platform. The live experiment is run continuously for two weeks. It involves an average of 25 million active users and billions of interactions each day. With such long time period and large scale of involved users, the recommendation environment can exhibit large deviations, as analysed in Sec. 3. This calls for the ability to adapt and explore in complex environments of online algorithms. To compare different algorithms, users are randomly split into several buckets. The first bucket runs the default RLUR algorithm, and the remaining buckets run models AdaRec, TD3, and ESCP.

The comparative results are shown in Tab. 1. The statistics are permillage or percentage improvements compared with RLUR. AdaRec exhibits superior performance in all evaluation metrics than baseline algorithms, including TD3, ESCP, and RLUR. Specifically, AdaRec is the only algorithm that achieves performance improvement in the application dwell time. AdaRec also improves the rate of user comments by 8.392‰, which is almost 5 times larger than the improvements of TD3. These empirical results demonstrate AdaRec's effectiveness and scalability when applied to real-world recommendation platforms.

Table 2: Results of ablation studies in the modified KuaiSim simulator.

| | Average Return Days (↓) | Return Probability @ Day 1 (↑) | Retention Reward (↑) |
|---|---|---|---|
| AdaRec (no exploration) | 1.868±0.061 | 0.708±0.015 | -0.018±0.000 |
| AdaRec (no encoder) | 1.803±0.109 | 0.732±0.034 | -0.018±0.001 |
| AdaRec (no distance-based loss) | 1.672±0.208 | 0.776±0.068 | -0.017±0.002 |
| AdaRec | **1.541**±0.056 | **0.819**±0.017 | **-0.015**±0.001 |

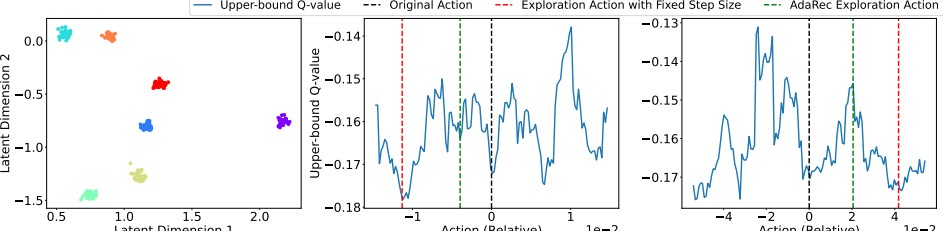

Figure 5: **Left:** Visualizations of the outputs of the context encoder in Sec. 4.1. States with different values are assigned with different colors. **Middle and Right:** The state-action value function on two example state-action pairs. The vertical lines show the relative values of the original action, the exploration action with a fixed step size, and the AdaRec exploration action.

## 5.4 ABLATIONS AND ANALYSES

We conduct ablation studies in the modified KuaiSim simulator to analyze the role of each module in AdaRec, namely the context encoder, the auxiliary loss to train the context encoder, and the selection of exploratory actions. As shown in Tab. 2, removing any of these components will lead to a drop in the overall algorithm's performance. Among them, the exploration module has the most significant impact. Without the safe and efficient exploration, the policy's performance is even lower than TD3 and RLUR as discussed in the previous section. This is because TD3 and RLUR have their respective exploration techniques. The comparison emphasizes the necessity of exploration in sequential recommendation tasks. We also analyse the computation cost of AdaRec's exploration module in Appendix D. The performance of AdaRec will also decrease without the context encoder or training the context encoder only with the policy loss (without the auxiliary loss related to $l_2$-distance in the latent space). This demonstrates the effectiveness of the context encoder and the loss function proposed in Sec. 4.1.

We visually demonstrate two of the key components of the AdaRec algorithm in Fig. 5. The left figure illustrates the outputs of the context encoders with a batch of states as input. The outputs are projected into two dimensions with PCA for visualization and colored according to the state values. As shown in the figure, states with similar values exhibit closely gathered encoder outputs , while those in different value categories tend to have distinct encoder outputs. In this way, the context encoder can help the learning policy identify whether it will perform well in the current environment. The middle and right figures show the landscapes of state-action values when the action is altered in one dimension. Our exploration policy can find a better action (vertical green line) than the exploration action generated with a fixed step size (vertical red line).

## 6 CONCLUSION

In this work, we address the challenge of distribution shift in optimizing long-term user engagement with sequential recommendation. Through data-driven analyses, we identify evolving patterns in user behavior, such as feedback frequencies and the return time distribution, as the main causes of distribution shift. To tackle this challenge, we propose the AdaRec algorithm for training adaptive recommendation policies. AdaRec utilizes a context encoder in the policy network that enables RL policies to identify different user behavior patterns. To facilitate fast policy adaptation, we combine the idea of optimism under uncertainty with zero-order optimization to boost exploration. Experimental results demonstrate that AdaRec outperforms state-of-the-art methods in optimising long-term user engagement and ensures stable recommendation quality in face of environment distribution shift.

**Ethics Statement**  While our approach involves real user logs, the user data we used have been stripped of all sensitive privacy information. Each user is denoted by an anonymous user-id, and sensitive features are encrypted.

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

# A  DETAILED DERIVATIONS

The loss $J_{\text{same}}(\phi))$ can be derived as follows:

$$
\begin{aligned}
J_{\text{same}}(\phi) &= \sum_{k=0}^{n} \sum_{s_i, s_j \in C_k} \|\phi(s_i) - \phi(s_j)\|_2^2 \\
&= \sum_{k=0}^{n} \left[ \sum_{s_i \in C_k} 2|C_k| \|\phi(s_i)\|_2^2 - 2 \sum_{s_i, s_j \in C_k} \phi^T(s_i)\phi(s_j) \right] \\
&= \frac{2B}{n} \sum_{k=0}^{n} \sum_{s_i \in C_k} \phi^T(s_i)(\phi(s_i) - \bar{\phi}_k) \\
&= \frac{2B}{n} \sum_{k=0}^{n} \sum_{s_i \in C_k} \phi^T(s_i)(\phi(s_i) - \bar{\phi}_k) - \bar{\phi}_k^T(\phi(s_i) - \bar{\phi}_k) \\
&= \frac{2B}{n} \sum_{k=0}^{n} \sum_{s_i \in C_k} \|\phi(s_i) - \bar{\phi}_k\|_2^2 .
\end{aligned} \tag{8}
$$

The loss $J_{\text{diff}}(\phi)$ can be derived as follows:

$$
\begin{aligned}
J_{\text{diff}}(\phi) &= - \sum_{k=0}^{n} \sum_{m>k} \sum_{s_i \in C_k} \sum_{s_j \in C_m} \|\phi(s_i) - \phi(s_j)\|_2^2 \\
&= - \sum_{k=0}^{n} \sum_{m>k} \sum_{s_i \in C_k} \left[ \frac{1}{|C_m|} \sum_{s_j \in C_m} \|\phi(s_i) - \phi(s_j)\|_2^2 \sum_{s_j \in C_m} 1^2 \right] \\
&\leqslant - \sum_{k=0}^{n} \sum_{m>k} \sum_{s_i \in C_k} \left[ \frac{n}{B} \left\| \sum_{s_j \in C_m} [\phi(s_i) - \phi(s_j)] \right\|_2^2 \right] \\
&\leqslant - \frac{B}{n} \sum_{k=0}^{n} \sum_{m>k} \sum_{s_i \in C_k} \|\phi(s_i) - \bar{\phi}_m\|_2^2 \\
&\leqslant - \frac{B^2}{n^2} \sum_{k=0}^{n} \sum_{m>k} \|\bar{\phi}_k - \bar{\phi}_m\|_2^2 .
\end{aligned} \tag{9}
$$

# B  ADDITIONAL RELATED WORK

## B.1  RECOMMENDATION WITH EVOLVING USER INTERESTS

Previous point-wise or list-wise recommendation models have incorporated the concept of evolving user behavior, primarily focusing on changes in the distribution of user interests (Steck, 2018; Kaya & Bridge, 2019). However, our paper considers other user behaviors related to the long-term user experience, such as the rate of immediate response and the distribution of user return time. To model the evolving user interests, Zhou et al. (2018; 2019) learn the representation of user interests from historical behaviors and performs click-through rate prediction. Brown & Agarwal (2022) proposes to ensure that sufficiently diversified content is recommended to the user in face of adaptive preferences. But the algorithm is analysed in the bandit setting without empirical justifications. Zhao et al. (2021) predicts users' shift in tastes during training and incorporates these predictions into a post-ranking network. Another approach involves predicting the distributions of the top-k target customers and training the recommendation model accordingly (Zhao et al., 2020). It is also worth noting that some studies have indicated that these forms of distribution adjustment may negatively impact the overall recommendation accuracy (Kleinberg et al., 2017; Zhao et al., 2020). Instead of predicting user behaviors and train recommendation models beforehand, our paper focus on the identification of new distributions and the ability to rapidly adapt to them.

Table 3: The hyperparameters for the AdaRec algorithm.

| | Hyperparameter | Value |
|---|---|---|
| Training | Optimizer | Adam |
| | Learning rate | $3 \cdot 10^{-4}$ |
| | Batch size $B$ | 256 |
| Deepnet | Number of transformer layers | 2 |
| | Dimension of feedforward networks | 64 |
| | Number of attention heads | 4 |
| RL | Discount factor $\gamma$ | 0.9 |
| | Replay buffer size | $5 \times 10^4$ |
| | Target smoothing coefficient | 0.005 |
| | Target update interval | 1 |
| AdaRec | Number of clusters $n$ | 4 |
| | $\beta$ in exploration | 30 |
| | $\delta$ in exploration | 1 |
| | Number of candidate actions | 6 |

## C ALGORITHM

The detailed algorithm procedure of AdaRec is shown in Alg. 1. The main differences between AdaRec and TD3 are: 1. The policy takes an additional latent variable $z_t$ as input (line 3). The latent variable is the output of the context encoder $\phi_\varphi$, which is trained with the loss specified in Eq. (6) (line 9). 2. The exploration action $a_E$ is generated with Eq. (7) (line 6) rather than by adding Gaussian noise to the original action $a_T$.

---

**Algorithm 1** The workflow of AdaRec

1: **Input:** The context encoder $\phi_\varphi$, the deterministic policy $\pi_\theta$, the state-action value function $Q_\psi$, the replay buffer $D$, training steps $N$, and the training horizon $H$.
2: Initialize the networks and the replay buffer.
3: **for** $1, 2, 3, \ldots, N$ **do**
4:   **for** $t = 1, 2, \ldots, H$ **do**
5:     Obtain $z_t$ from $\phi_\varphi(s_t)$ and then sample $a_T$ from $\pi_\theta(s_t, z_t)$.
6:     Get the exploration action $a_E$ with Eq. (7) and set $a_t = a_E$.
7:     Interact with the simulator, get transition data $(s_{t+1}, r_t, d_{t+1}, s_t, a_t, z_t)$, and add it to $D$.
8:   **end for**
9:   Update the context encoder $\phi_\varphi$ according to Eq. (6).
10:   Use the replay buffer $D$ and the TD3 (Fujimoto et al., 2018) algorithm to update the policy and value network parameters $\theta$ and $\psi$.
11: **end for**

---

## D ADDITIONAL EXPERIMENT DETAILS

### D.1 SETUP

The network architecture of the retention simulator is simular to the policy network. We assume the immediate user response follows a Bernoulli distribution and the user return time in days follows a geometric distribution. The simulator is trained in a style of supervised learning and is updated by likelihood maximization on the training data. The hyperparameters for the AdaRec algorithm during training are specified in Tab. 3.

### D.2 DETAILS OF BASELINE ALGORITHMS

We consider the following algorithms as baseline methods:

- TD3 (Fujimoto et al., 2018): A value-based off-policy RL algorithm that incorporates a pair of Q-networks to mitigate overestimation.

- SAC (Haarnoja et al., 2018): A value-based off-policy RL algorithm with a stochastic policy and the maximum-entropy RL objective.

- OAC (Ciosek et al., 2019): An RL-based exploration algorithm that incorporates the idea of optimism-under-uncertainty, and obtains a separate optimistic action during the training phase.

- RND (Burda et al., 2019): An RL-based exploration algorithm that encodes the state input to fit the output of a random network. States with larger encoding error will be assigned with higher intrinsic reward.

- ESCP (Luo et al., 2022): A typical environment sensitive contextual Meta-RL approach that explicitly identifies environment parameters as additional policy inputs.

- RLUR (Cai et al., 2023a): An RL-based recommendation algorithm especially designed for optimizing long-term user engagement.

- CEM (Deng, 2006): The Cross Entropy Method, which is commonly used as a surrogate for RL algorithms in recommendation tasks.

- DIN (Zhou et al., 2019): The Deep Interest Network that learns the representation of user interests from historical behaviors and performs click-through rate prediction.

### D.3 TABLE FOR FINAL PERFORMANCE COMPARISONS

We show the final performance comparisons of AdaRec and all baseline algorithms in Tab. 4. The exploration algorithms OAC and RND show inferior performance due to the unprotected action selection mechanism. One improper choice of recommendation item risks boring the user and terminating the whole episode. DIN also performs worse than AdaRec, in that supervised-learning algorithms can only learn from the immediate response, which is the click-through rate in DIN's formulation. Instead, AdaRec is a RL algorithm that has the ability to capture the long-term effect of the recommended items and is more suitable for optimizing long-term user engagement.

Table 4: The final performance comparisons of AdaRec and all the baseline algorithms. The scores are computed at the final timestep (50K) of training.

|  | Average Return Days ($\downarrow$) | Return Probability at Day 1 ($\uparrow$) | Reward ($\uparrow$) |
|---|---|---|---|
| CEM | 1.841±0.214 | 0.720±0.067 | -0.017±0.001 |
| DIN | 1.725±0.029 | 0.755±0.005 | -0.017±0.001 |
| TD3 | 2.023±0.012 | 0.659±0.002 | -0.020±0.000 |
| SAC | 2.023±0.012 | 0.659±0.002 | -0.020±0.000 |
| OAC | 1.778±0.122 | 0.738±0.040 | -0.017±0.001 |
| RND | 1.704±0.131 | 0.765±0.045 | -0.016±0.001 |
| ESCP | 1.719±0.098 | 0.759±0.032 | -0.017±0.000 |
| RLUR | 1.910±0.066 | 0.693±0.019 | -0.019±0.000 |
| AdaRec (Ours) | **1.541**±0.056 | **0.819**±0.017 | **-0.015**± 0.000 |

### D.4 COMPUTATION COST OF THE EXPLORATION MODULE

AdaRec requires addtional steps in action selection, computing the gradient of the Q-function and sampling among candidate actions. But apart from action selection, RL training involves interacting with the environment and updating the policy with gradient decent. These two parts will take up more time than action selection. We conduct empirical studies and exhibit in Tab. 5 the average time cost of action selection in one training step. The total time cost of one training step is also shown for comparison. According to the results, although the exploration module lead to an addtional 129% of computation cost, it only costs less than 10% more total time. Also, during deployment the exploration module is not included, so it adds no more computation cost.

Table 5: Average time cost of action selection in one training step.

|  | Action Selection (s) | Total Time (s) |
|---|---|---|
| AdaRec | 0.259 | 1.738 |
| AdaRec (no exploration) | 0.113 | 1.595 |
| Exploration Time Cost | 129 % | 8.966 % |

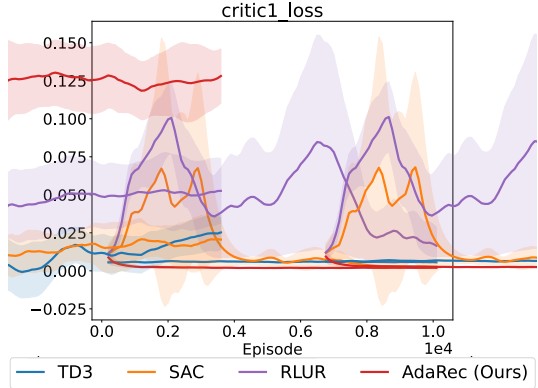

Figure 6: The critic training loss of selected value-based algorithms.

## D.5 CURVE FOR CRITIC LOSS

We also exhibit the curve of the value function training loss in Fig. 6. Thanks to the additional encoder output $\phi(s)$ as input of the value function, AdaRec has the lowest and most stable critic loss among selected value-based algorithms. TD3 has a higher but stable critic loss mainly because of its insufficient exploration. Although SAC and RLUR have their respective exploration modules, they cannot adapt to environment changes and have an unstable critic loss curve.

