# OpenReview forum: "AdaRec: Adaptive Sequential Recommendation for Reinforcing Long-term User Engagement"
_ICLR.cc/2024/Conference — Submitted to ICLR 2024_

### Official Review · Reviewer_TqZE · 2023-10-29

**Soundness:** 2 fair
**Presentation:** 2 fair
**Contribution:** 2 fair
**Rating:** 3
**Confidence:** 4

**Summary:**

This paper studied the challenges of users’ complicated behavior changes in online recommendation systems. In different periods, users’ behaviors change in terms of preferences, return time and frequencies of immediate user feedback. To handle this challenge, the authors propose an adaptive sequential recommendation method to optimize long-term user engagement. Specifically, the authors utilize a context encoder to encode user’s behavior patterns and regularize the encoder to produce a similar latent representation for states with similar state values. An optimistic exploration is further utilized to encourage exploration. Experiments are carried out using a recommender simulator and an online A/B test.

**Strengths:**

1.	The paper is well-organized and easy to follow.
2.	Experiments are carried out on both a recommender simulator and an online A/B test, which is comprehensive.
3.	The authors conduct an ablation study to validate the effectiveness of each component.

**Weaknesses:**

1.	One major concern about the proposed method is the regularization loss in the proposed context encoder, which seems problematic to me. The motivation of this paper is to handle the distribution shift of the evolving user behavior patterns. However, encouraging states with similar state values to have similar latent encoding representation does not solve the distribution shift issues. The estimated state value function can still face the challenge of user behavior shift, resulting in an inaccurate state value estimation.

2.	Another major concern is the novelty of the proposed method. To my knowledge, using context encoder to encoder user behavior patterns is not new in the recommendation context, which is also discussed in the related work section. The novelty of adding regularization loss in the context encoder is limited. The adopted exploration mechanism from RL literature is rather general and it is unclear how it particularly handles the user exploration in the recommendation context, which usually involves large action space.

3.	As this paper aims to handle the user behavior evolution challenge in the sequential recommendation setting, some baselines in the Non-RL recommender literature are missing such as [1, 2].

[1] Zhou, G., Zhu, X., Song, C., Fan, Y., Zhu, H., Ma, X., ... & Gai, K. (2018, July). Deep interest network for click-through rate prediction. In Proceedings of the 24th ACM SIGKDD international conference on knowledge discovery & data mining (pp. 1059-1068).

[2] Zhou, G., Mou, N., Fan, Y., Pi, Q., Bian, W., Zhou, C., ... & Gai, K. (2019, July). Deep interest evolution network for click-through rate prediction. In Proceedings of the AAAI conference on artificial intelligence (Vol. 33, No. 01, pp. 5941-5948).

**Questions:**

See the Weaknesses for the questions.

---

> ### Author Response · Authors · 2023-11-14
> **Rebuttal for Reviewer TqZE (Part I)**
>
> Thank you for your insightful comments. We provide discussions on your concerns as follows.
>
> Q: **One major concern about the proposed method is the regularization loss in the proposed context encoder, which seems problematic to me. The estimated state value function can still face the challenge of user behavior shift, resulting in an inaccurate state value estimation.**
>
> A: As demonstrated in Figure 2, the encoder output $\phi(s)$ is also used as the input to the state-value function $V$ for more accurate value estimation. In Equation (2) the input $\phi(s)$ of the $V$-function is omitted for brevity. We add the clarification in Section 4.1 of the revised paper.  To be specific, the optimization of the context encoder and the value functions $V$ follows an iterative procedure. Since the distance-based encoder loss do not directly rely on the accrate output of $V$ (it only uses $V$ to rank the states and assign them into groups),  value functions on the previous timestep is enough to generate an accurate encoder loss.
> We also exhibit the curve of the value function training loss in Appendix D.5. The numerical results are posted here for your reference.
>
> | Algorithm | TD3 | SAC | RLUR | AdaRec (Ours) |
> | :--- | :---: | :---: | :---: | :---: |
> | Critic loss on 10k steps | 6.88e-3 | 2.81e-3 | 2.52e-3 | **1.85e-3** |
>
> Thanks to the encoder output $\phi(s)$ as additional input of the value function, AdaRec has the lowest and most stable critic loss among selected value-based algorithms. This means that AdaRec has a relatively accurate value estimation during training. TD3 has a higher but stable critic loss mainly because of its insufficient exploration. Although SAC and RLUR have their respective exploration modules, they cannot adapt to environment changes and have an unstable critic loss curve.
>
> Q: **Another major concern is the novelty of the proposed method.  The novelty of adding regularization loss in the context encoder is limited. It is also unclear how the exploration mechanism handles the user exploration in the recommendation context, which usually involves large action space.**
>
> A: Context encoders themselves are only one of the network architechtures that enable more delicated design of state representations. It is the loss functions that differentiate encoder designs. There are many peer-reviewed RL papers that solely focus on proposing new losses for training the context encoders [1,2,3,4]. With regard to our paper, the major concern for designing encoder loss is that a variety of latent factors, including user behavior patterns, user preferences, and users' return time distributions, can collaboratively influence how the environment change. Previous methods that attempt to recover one single latent factor with context encoders will fail in this scenario. Our AdaRec algorithm manages to relate the encoder output with the estimated long-term return, which serves as a universal criterion to detect whether the environment changes: the policy will obtain low rewards with distribution shifts. The performance comparison between ESCP, which is the state-of-the-art context encoder-based algorithm, and AdaRec in both the simulator and the live experiments also demonstrates the effectiveness of our loss for training the encoder.
>
> With respect to the exploration mechanism, we would like to emphasize that instead of setting the action space to be all candidate items, we use a 7-dimentional continuous action space for RL training. In this setting the action space is not large. The actions are combined with a deep scoring model and generate a ranking score for each candidate items. Please kindly check the updated Section 2.1 for details. In this problem formulation, the major concern is that recommendation tasks can be risk-sensitive: users may disengage from the application and cease the recommendation process if they encounter recommended items that fail to capture their interest. So relying solely on optimistic actions will be harmful to the recommendation quality. In AdaRec, the zero-order action selection procedure guarantees that potentially undesired actions will not lead to improper recommendation.

---

> ### Author Response · Authors · 2023-11-14
> **Rebuttal for Reviewer TqZE (Part II)**
>
> Thank you for your insightful comments. We provide discussions on your concerns as follows.
>
> Q: **Some baselines in the Non-RL recommender literature are missing.**
>
> A: We thank the reviewers for pointing it out. The mentioned papers are discussed in the related work section in Appendix B.1. We also implement DIN and compare its performance with AdaRec. We keep the original input of DIN unchanged and use transformers, which has shown better performance than vanilla fully-connected networks, as the backbone network. This help us make fair comparisons. The comparative results are in Appendix D.3 of the revised paper. We also list them here for your reference.
>
> |  | Average Return Days ( $\downarrow$ ) | Return Probability at Day 1 $(\uparrow)$ | Reward $(\uparrow)$ |
> | :--- | :---: | :---: | :---: |
> | CEM | $1.841 \pm 0.214$ | $0.720 \pm 0.067$ | $-0.017 \pm 0.001$ |
> | DIN | $1.725 \pm 0.029$ | $0.755 \pm 0.005$ | $-0.017 \pm 0.001$ |
> | TD3 | $2.023 \pm 0.012$ | $0.659 \pm 0.002$ | $-0.020 \pm 0.000$ |
> | SAC | $2.023 \pm 0.012$ | $0.659 \pm 0.002$ | $-0.020 \pm 0.000$ |
> | OAC | $1.778 \pm 0.122$ | $0.738 \pm 0.040$ | $-0.017 \pm 0.001$ |
> | RND | $1.704 \pm 0.131$ | $0.765 \pm 0.045$ | $-0.016 \pm 0.001$ |
> | ESCP | $1.719 \pm 0.098$ | $0.759 \pm 0.032$ | $-0.017 \pm 0.000$ |
> | RLUR | $1.910 \pm 0.066$ | $0.693 \pm 0.019$ | $-0.019 \pm 0.000$ |
> | AdaRec (Ours) | $\mathbf{1.541} \pm 0.056$ | $\mathbf{0.819} \pm 0.017$ | $\mathbf{- 0.015} \pm 0.000$ |
>
> As shown in the reults, DIN performs worse than AdaRec. This is because supervised-learning algorithms can only learn from the immediate response, which is the click-through rate in DIN's formulation. Instead, AdaRec is a RL algorithm that has the ability to capture the long-term effect of the recommended items and is more suitable for optimizing long-term user engagement.
>
> [1] Context-aware Dynamics Model for Generalization in Model-Based Reinforcement Learning. ICML 2020.
>
> [2] Trajectory-wise Multiple Choice Learning for Dynamics Generalization in Reinforcement Learning. NeurIPS 2020.
>
> [3] Learning Invariant Representations for Reinforcement Learning without Reconstruction. ICLR 2021.
>
> [4] Adapt to Environment Sudden Changes by Learning a Context Sensitive Policy. AAAI 2022.

---

> ### Author Response · Authors · 2023-11-17
> **Replying to Reviewer TqZE**
>
> Dear Reviewer TqZE,
>
> We sincerely value your dedicated guidance in helping us enhance our work. We are eager to ascertain whether our responses adequately address your primary concerns, particularly in relation to the effectiveness of the context encoder, the novelty of the proposed approach, and performance comparison with non-rl recommender algorithms. Thank you.

---

> ### Author Response · Authors · 2023-11-20
> **Replying to Reviewer TqZE**
>
> Dear Reviewer TqZE,
>
> As the author-reviewer discussion period is approaching the end, we again extend our gratitude for your reviews that help us improve our paper. Any further comments on our rebuttal or the revised paper are appreciated. Thanks for your time!

---

### Official Review · Reviewer_dRAr · 2023-11-01

**Soundness:** 2 fair
**Presentation:** 2 fair
**Contribution:** 3 good
**Rating:** 6
**Confidence:** 3

**Summary:**

The paper introduces a novel paradigm called AdaRec to address the challenge of evolving user behavior patterns in large-scale online recommendation systems. The goal is to optimize long-term user engagement by leveraging Reinforcement Learning (RL) algorithms. By introducing a distance-based representation loss to extract latent information from users' interaction trajectories, AdaRec helps the RL policy identify subtle changes in the recommendation system. To enable rapid adaptation, AdaRec encourages exploration using the idea of optimism under uncertainty. It also incorporates zero-order action optimization to ensure stable recommendation quality in complex environments.

**Strengths:**

1. The problems studied in this paper exist widely in recommendation systems, and have been ignored by previous researchers, which is a very promising and important research direction.
2. The paper presents a distance-based representation loss to identify the subtle user behavior patterns changes, which is novel and interesting.
3. Extensive empirical analyses in simulator-based and live sequential recommendation tasks demonstrates that AdaRec outperforms baseline algorithms in terms of long-term performance.

**Weaknesses:**

1. The writing of the paper needs further improvement.
(1)	What is the specific meaning of State Space S?
(2)	The paper should give a brief introduction before using some reinforcement learning concepts.
2. Although I agree with that the user behavior patterns are evolving in recommendation systems, the study in Section 3 does not make sense to me. At different time stamp, user interaction history statistics are different, that is, the distribution of states are different, so different interaction frequency may not reflect the change of user behavior patterns. There is no evidence to support the argument “As previously discussed, given the same distribution of states s_t and actions a_t, the users’ return time exhibits fluctuations across different weeks.”
3. In reinforcement learning, there are many exploration strategies. What are the advantages of the schemes mentioned in the paper, and the comparison experiments with other schemes need to be provided.

**Questions:**

1. What is the specific meaning of State Space S?
2. The comparison experiments with other exploration schemes need to be provided.

---

> ### Author Response · Authors · 2023-11-13
> **Rebuttal for Reviewer dRAr**
>
> Thank you for your insightful comments. We provide discussions on your concerns as follows.
>
> Q: **What is the specific meaning of State Space S? The paper should give a brief introduction before using some reinforcement learning concepts.**
>
> A: The State Space $\mathcal{S}$ refers to a set of all possible states that may occur in the trajectory of an MDP. In Section 2.1, we add RL concepts such as the accumulated return $\eta_T$, the state-action value function $Q_T^\pi(s, a)$, and the state value function $V_T^\pi(s)$. Please check the revised paper for details. We hope the additional RL concepts can make readers easier to understand our proposed algorithm.
>
> Q: **The study in Section 3 does not make sense to me. The distribution of states are different, so different interaction frequency may not reflect the change of user behavior patterns.**
>
> A: We thank the reviewer for pointing it out. To remove the influence of various interaction histories, we select interactions that happen at the start of each recommendation session, i.e., the $s_0$ in the trajectory. We exhibit the updated analysis results in Figure 1 (lower right) of the revised paper. According to the analysis, the user interaction frequency still exhibits large variations in different time periods. The mode of changes in user behavior patterns is similar to those in the original analysis. The return distribution in Figure 1 (upper right) is computed trajectory-wise, and depends on $(s_0,a_0,a_1,\cdots,a_n,\cdots)$ without interaction history. Its evolvement can also reflect the changes in user behavior patterns. We believe the updated data-driven study can better demonstrate the existence of distribution shift in recommender systems.
>
> Q: **In reinforcement learning, there are many exploration strategies. What are the advantages of the schemes mentioned in the paper? The comparison experiments with other exploration schemes also need to be provided.**
>
> A: Among exploration strategies in RL, there are two major mechanisms that are relatively simple and suitable for deployment in real-world applications. One is the optimism under uncertainty [1,2]. It encourages the learning agent to choose optimistically when it is uncertain about the outcome of certain actions. However, as clarified in Section 4.2, recommendation tasks can be risk-sensitive: users may disengage from the application and cease the recommendation process if they encounter recommended items that fail to capture their interest. So relying solely on optimistic actions will be harmful to the recommendation quality. In AdaRec, the exploration module is also based on the idea of optimism under uncertainty. But the zero-order action selection procedure guarantees that potentially undesired actions will not lead to improper recommendation.
>
> Another mechanism is curiosity-driven exploration [3]. It usually involves a random encoding network $N$ that maps the state $s$ into low-dimentional vectors $z$. Then another separate encoding network $N^{\prime}$ is trained to reconstruct the mapping from $s$ to $z$. If the visitation count to the state $s$ is small, the reconstruction loss of $N^{\prime}$ on state $s$ will be relatively large and $s$ will be more likely to be explored. But in our setting of adaptive recommendation, the outcome of the same state $s$ can be different with different user behavior patterns. So even when $s$ has been visited for many times, it is possible that with new behavior patterns the state $s$ still requires considerate exploration. Instead, the exploration mechanism in AdaRec relies on the optimistic Q-values that are more accurate than intuitive visitation counts.
>
> For comparative analysis, we consider OAC [1] and RND [3] as additional baselines. These two algorithms are representative approaches of the two exploration mechanisms respectively. We add the results in the revised paper, and also post them here for your reference:
>
> |  | Average Return Days $(\downarrow)$ | Return Probability at Day 1 $(\uparrow)$ | Reward $(\uparrow)$ |
> | :--- | :---: | :---: | :---: |
> | CEM | $1.841 \pm 0.214$ | $0.720 \pm 0.067$ | $-0.017 \pm 0.001$ |
> | TD3 | $2.023 \pm 0.012$ | $0.659 \pm 0.002$ | $-0.020 \pm 0.000$ |
> | SAC | $2.023 \pm 0.012$ | $0.659 \pm 0.002$ | $-0.020 \pm 0.000$ |
> | OAC | $1.778 \pm 0.122$ | $0.738 \pm 0.040$ | $-0.017 \pm 0.001$ |
> | RND | $1.704 \pm 0.131$ | $0.765 \pm 0.045$ | $-0.016 \pm 0.001$ |
> | ESCP | $1.719 \pm 0.098$ | $0.759 \pm 0.032$ | $-0.017 \pm 0.000$ |
> | RLUR | $1.910 \pm 0.066$ | $0.693 \pm 0.019$ | $-0.019 \pm 0.000$ |
> | AdaRec (Ours) | $\mathbf{1.541} \pm 0.056$ | $\mathbf{0.819} \pm 0.017$ | $\mathbf{-0.015} \pm 0.000$ |
>
> Our AdaRec algorithm can outperform OAC and RND in all metrics, demonstrating its superior ability to make safe explorations.
>
> [1] Better Exploration with Optimistic Actor-Critic.
>
> [2] Seizing Serendipity: Exploiting the Value of Past Success in Off-Policy Actor-Critic.
>
> [3] Exploration by Random Network Distillation.

---

> ### Author Response · Authors · 2023-11-17
> **Replying to Reviewer dRAr**
>
> Dear Reviewer dRAr,
>
> We sincerely value your dedicated guidance in helping us enhance our work. We are eager to ascertain whether our responses adequately address your primary concerns, particularly in relation to the new empirical analysis in Section 3 and the additional performance comparison with other exploration schemes in the experiments.

---

> ### Author Response · Authors · 2023-11-20
> **Replying to Reviewer dRAr**
>
> Dear Reviewer dRAr,
>
> We noticed that you have raised the score to 6. Thank you for the acknowledgement!

---

### Official Review · Reviewer_kKUq · 2023-11-04

**Soundness:** 3 good
**Presentation:** 3 good
**Contribution:** 3 good
**Rating:** 6
**Confidence:** 3

**Summary:**

The paper presents a novel paradigm to tackle the challenge of distribution shift in large-scale online recommendation systems. In these systems, the dynamics and reward functions are continuously affected by changes in user behavior patterns, making it difficult for existing reinforcement learning (RL) algorithms to adapt effectively.

AdaRec proposes a multi-faceted approach to address this issue. It introduces a distance-based representation loss, which extracts latent information from users' interaction trajectories. This information reflects how well the RL policy aligns with current user behavior patterns, allowing the policy to detect subtle changes in the recommendation system.

AdaRec's approach to addressing distribution shift in recommendation systems appears promising and aligns with current challenges in the field. The full paper's empirical results and detailed methodology will be necessary to assess the significance and practical applicability of this novel paradigm in real-world recommendation systems.

**Strengths:**

1. AdaRec introduces a novel paradigm for addressing distribution shift in large-scale online recommendation systems, which is a significant and challenging problem in the field.

2. The use of zero-order action optimization to ensure stable recommendation quality in complicated environments is a strong point, as it addresses the need for robustness in real-world recommendation systems.

3. The claim of superior long-term performance is supported by extensive empirical analyses in both simulator-based and live sequential recommendation tasks, indicating a commitment to evaluating the proposed solution rigorously.

**Weaknesses:**

1. The use of "optimism under uncertainty" and zero-order action optimization may introduce additional complexity to the approach, which could be a drawback in terms of implementation and computational cost.

**Questions:**

Does the computational cost of reinforcement learning need to be analyzed?

---

> ### Author Response · Authors · 2023-11-13
> **Rebuttal for Reviewer kKUq**
>
> Thank you for your insightful comments. We provide discussions on your concerns as follows.
>
> Q: **The use of "optimism under uncertainty" and zero-order action optimization may introduce additional complexity to the approach, which could be a drawback in terms of implementation and computational cost.**
>
> A: In terms of implementation, the major concern may be the gradient $\left[\nabla_a Q_{\mathrm{UB}}(s, a)\right]_{a=a_T}$. A similar form of gradient is computed when updating the Q-networks, so the gradient computation will not lead to heavy implementation cost.
>
> In terms of computational cost, AdaRec requires addtional steps in action selection, computing the gradient of the Q-function and sampling among candidate actions. But apart from action selection, RL training involves interacting with the environment and updating the policy with gradient decent. These two parts will take up more time than action selection. We conduct empirical studies and exhibit the average time cost of action selection in one training step. The total time cost of one training step is also shown for comparison. According to the results, although the exploration module lead to an addtional 129\% of computation cost, it only costs less than 10\% more total time. Also, during deployment the exploration module is not included, so it adds no more computation cost.
>
> |  | Action Selection (s) | Total Time (s) |
> | :--- | :---: | :---: |
> | AdaRec | 0.259 | 1.738 |
> | AdaRec (no exploration) | 0.113 | 1.595 |
> | Exploration Time Cost | 129 \% | 8.966 \% |
>
> The discussions are added to Appendix D of the revised paper.

---

> > ### Comment · Reviewer_kKUq · 2023-11-23
> > **Official Comment by Reviewer kKUq**
> >
> > Thanks for the detailed response. I have read the rebuttal and most of my concerns have been well addressed. Overall, I am towards acceptance and will maintain my score.
> >
> > Best, The reviewer kKUq

---

### Meta-Review · Area_Chair_aNmL · 2023-12-14

**Metareview:**

This paper presents a new paradigm to tackle distribution shift in large-scale online recommendation systems, where dynamics and reward functions are continuously affected by changes in user behavior patterns. The authors utilize a context encoder to encode user’s behavior patterns and regularize the encoder to produce a similar latent representation for states with similar state values. Experiments are carried out using simulated environments and online A/B tests.

Strengths: the paper is well organized and easy to follow. The authors conducted extensive experiments in simulated environments and online A/B tests.
Weaknesses: I share similar concerns as reviewer TqZE. If the authors assumed the distribution shift is completely captured by change in the user behavior patterns, then context encoder itself should already capture the change, and using context encoder to encode historical user behavior pattern is not new in RecSys. On the other hand, if the authors assumed there are exogenous factors influenced the dynamics and reward, then the estimated state functions have errors as well, and thus regularization loss using the closeness of the state value estimation is problematic.

**Justification For Why Not Higher Score:**

I share similar concerns as reviewer TqZE. If the authors assumed the distribution shift is completely captured by changes in the user behavior patterns, then context encoder itself should already capture the change, and using context encoder to encode historical user behavior pattern is not new in RecSys. On the other hand, if the authors assumed there are exogenous factors that influenced the dynamics and reward, then the estimated state functions have errors as well, and thus employing a regularization loss using the closeness of the state value estimation is problematic.

**Justification For Why Not Lower Score:**

N/A

---

### Decision · Program_Chairs · 2024-01-16

Reject